# Visit-to-visit blood pressure variability is common in primary care patients: Retrospective cohort study of 221,803 adults

Finlay A. McAlister[1]*, Brendan Cord Lethebe[2], Alexander A. Leung[3], Rajdeep S. Padwal[1], Tyler Williamson[2,4]

**1** Division of General Internal Medicine, Faculty of Medicine & Dentistry, University of Alberta, Edmonton, Canada, **2** Clinical Research Unit, Cumming School of Medicine, University of Calgary, Calgary, Canada, **3** Department of Medicine, Cumming School of Medicine, University of Calgary, Calgary, Canada, **4** Division of Community Medicine, Cumming School of Medicine, University of Calgary, Calgary, Canada

* Finlay.McAlister@ualberta.ca

## Abstract

**Data Availability Statement:** Although data sharing agreements between individual practitioners and the Canadian Primary Care Sentinel Surveillance Network (CPCSSN) prohibit

### Objective

Although high visit-to-visit blood pressure variability (BPV) is an independent risk factor for cardiovascular events, the frequency of high BPV is unknown. We conducted this study to define the frequency of high BPV in primary care patients, clinical correlates, and association with antihypertensive therapies.

### Methods

Retrospective cohort study using electronic medical record data (with previously validated case definitions based on billing codes, free text analysis of progress notes, and prescribing data) from the Canadian Primary Care Sentinel Surveillance Network of 221,803 adults with multiple clinic visits over a 2-year period. We *a priori* defined a standard deviation>13.0 mm Hg in visit-to-visit systolic blood pressure (SBP) as "high BPV" based on prior literature.

### Results

Overall, 85,455 (38.5%) patients had hypertension (mean 6.56 visits with SBP measurement, mean SBP 134.4 with Standard Deviation [SD] 11.3, 33.2% exhibited high BPV) and 136,348 did not (mean 3.96 visits with SBP measurement, mean SBP 120.9 with SD 8.2, 16.5% had high BPV). BPV increased with age regardless of whether individuals had hypertension or not; at all ages BPV varied across antihypertensive treatment regimens and was greater in those receiving renin angiotensin blockers or beta-blockers (p<0.001). High BPV was more frequent in patients with diabetes, chronic kidney disease, dementia, depression, chronic obstructive pulmonary disease, or Parkinson's disease.

us from making the dataset publicly available since it contains potentially sensitive health information (a restriction imposed by the Queen's University Health Research Ethics Board for any research use of the CPCSSN data), access may be granted to those who meet pre-specified criteria for confidential access, available by contacting info@cpcssn.org. The Canadian Primary Care Sentinel Surveillance Network (CPCSSN) has a formal data sharing policy (see http://cpcssn.ca/join-cpcssn/for-researchers/). Data will be disclosed only upon request and approval of the proposed use of the data by a review committee created by leaders of the network. This review will serve to ensure that patient privacy and rights, and data and research integrity, can be maintained. Review criteria will include demonstrated competence in data security and analysis and data will be shared to achieve the objectives in the approved protocol only. Anonymized data and a data dictionary will be made available, subject to requirements or restrictions from research ethics board or institutional review boards, existing contracts or agreements, and conditions set forth in participant consent forms. Data will be made available through secure data transfer methods overseen by CPCSSN and Queen's University, Kingston, Ontario Canada, or by having analyses performed by the CPCSSN Staff, subject to capacity. Each proposal must identify and provide funding to defray the costs of data preparation, storage, transfer, and analysis for the organization incurring these costs.

**Funding:** No project specific funding but FAM is supported by the Alberta Health Services Chair in Cardiovascular Outcomes Research.

**Competing interests:** No authors have competing interests.

## Conclusions

High visit-to-visit BPV is present in one sixth of non-hypertensive adults and one third of hypertensive individuals and is more common in those with comorbidities. The frequency of high BPV varies across antihypertensive treatment regimens.

## Introduction

Despite emerging evidence that within-patient visit-to-visit blood pressure variability (BPV) is an independent risk factor for hypertensive target organ damage and cardiovascular events, this phenomenon is under-appreciated by clinicians and discrepant readings at serial clinic visits are frequently dismissed as random fluctuations [1,2]. It is unclear whether the excess cardiovascular risk associated with BPV is a direct result of the mechanical effect of blood pressure variations or merely reflects the fact that the abnormal cardiovascular regulatory mechanisms that lead to higher visit-to-visit BPV also cause higher CV event rates [3–9].

A systematic review of antihypertensive trials suggested that treatment with calcium channel blockers or thiazide diuretics reduced BPV while angiotensin converting enzyme inhibitor (ACEi), angiotensin receptor blocker (ARB), and beta-blocker therapies were all associated with increases in BPV [10]. A subsequent secondary analysis of the Antihypertensive and Lipid-Lowering Treatment to Prevent Heart Attack Trial data confirmed that BPV was higher in the lisinopril treated patients compared to those treated with chlorthalidone or amlodipine [11].

While prior publications have established that high BPV is a risk factor for subsequent cardiovascular events [4–9], they were unable to quantify the prevalence of this risk factor since definitions were based on classifying BPV within studies on the basis of quartiles, quintiles, or deciles of SD. However, perusal of the actual SD values in those publications led us to *a priori* define a SD > 13.0 mm Hg in visit-to-visit SBP as "high BPV" [1–9] and to use that value with the aim of examining how common high BPV is in primary care practice and whether it differs across comorbidities and treatment regimens using data from the Canadian Primary Care Sentinel Surveillance Network (CPCSSN) [12].

## Methods

As outlined in detail elsewhere [12], the CPCSSN is a pan-Canadian electronic medical record (EMR) database with central collation and cleaning of data which includes approximately 1400 primary care physicians and nurse practitioners and approximately 1.8 million patients. CPCSSN data includes patient demographics, physician-assigned diagnoses (both as free text and *International Classification of Diseases, Ninth Revision* codes), physical measurements (including blood pressure), prescriptions, and laboratory results. For the purposes of this retrospective cohort study we included all adults who had at least two visits to a CPCSSN clinic between June 1, 2013 and June 1, 2015. Ethics approval was obtained from the Conjoint Health Research Ethics Board at the University of Calgary with waiver of individual patient consent.

We used a previously validated EMR-based case definition [13] with 85% sensitivity and 94% specificity to identify patients with hypertension based on ICD-9CM billing codes, text words in progress notes, and prescriptions for antihypertensive drugs. We used a combination of text word searching and ICD-9CM billing codes to identify comorbidities using case definitions previously validated in CPCSSN (we included fasting glucose, HbA1C, and prescriptions

for glucose lowering medications in the case definition for diabetes mellitus and estimated glomerular filtration rate <60 mL/min/1.73 m$^2$ in the case definition for chronic kidney disease) [12–14]. We defined antihypertensive drugs used by each patient on the basis of prescriptions dispensed within the CPCSSN EMR and classified them using the ATC system (combining C08 (calcium channel blockers [CCB]) OR C03 (diuretics) and comparing those to C07 (beta-blockers [BB]) OR C09 (angiotensin converting enzyme inhibitors [ACEi] or angiotensin receptor blockers [ARB]) OR CO2A, CO2B, C02C (anti-adrenergic agents). Systolic Blood Pressure (SBP) measurements were extracted from the EMRs. Information on educational level, income, smoking, alcohol use was not recorded in most charts and we did not extract data on conditions like stroke, heart failure, or angina for which there are no validated case definitions yet within CPCSSN [13].

### Statistical analyses

The majority of the statistics were descriptive in nature. Categorical variables were described with percentages, while continuous variables were reported with means and standard deviations (SD). We defined visit-to-visit variability of SBP within each patient using the SD between the average SBP for that patient at each visit (BP$_i$) and all visits ($\underline{BP}$) as per other studies [6].

$$SD = \sqrt{\frac{\sum (BP_i - \bar{BP})^2}{n-1}}$$

Characteristics were compared between groups using Student's t-test (for continuous variables) or chi-square tests (for categorical variables) as appropriate. Data analyses were conducted with SAS Version 9.4 (SAS Institute, Cary, NC, USA) and differences were considered to be statistically significant when p-values were less than 0.05.

## Results

Of the 221,803 adults with at least 2 primary care visits over the 2 years we studied, 85,455 (38.5%) met our validated case definition of hypertension: 68,767 (31.0%) were known to have hypertension prior to the study period and 16,688 (7.5%) were newly diagnosed during the period studied. Patients with hypertension were older and had more comorbidities than those without a diagnosis of hypertension (all p<0.001, Table 1). Hypertensive individuals also had more SBP measurements than those without a diagnosis of hypertension (mean 6.56 vs. 3.96 over the 2 years, p<0.001). Patients with a diagnosis of hypertension had a mean SBP of 134.4 mm Hg with SD of 11.3 while the 136,348 patients without a diagnosis of hypertension exhibited a mean SBP of 120.9 mm Hg with SD 8.2 (p<0.001, Table 1). High BPV (SD>13.0 mm Hg) was more than twice as common in those with hypertension (33.2%) compared to those without hypertension (16.5%, p<0.001, Fig 1).

The highest quartile of inter-visit BPV was SD above 11.2 mm Hg in normotensive individuals and 14.5 in hypertensive people. BPV increased with age regardless of whether individuals had hypertension or not (Fig 2) and at all ages was more pronounced in patients treated with RAS and/or BB drugs, especially if they received additional agents (Fig 3, p<0.001). Patients with high BPV were older, had higher SBP measurements, and were more likely to have diabetes, chronic kidney disease, chronic obstructive pulmonary disease, dementia, depression, or Parkinson's disease (Table 1). The proportion of patients with high BPV was similar across the 10 CPCSSN centres contributing data to this analysis.

**Table 1. Patient characteristics.**

| | Patients with Hypertension (n = 85,455) | | P value for High BPV vs. not | Patients without Hypertension (n = 136,348) | | P value for High BPV vs. not | P value for hypertension vs. not |
|---|---|---|---|---|---|---|---|
| | High BPV n = 28343 | Non-high BPV n = 57112 | | High BPV n = 22472 | Non-high BPV n = 113876 | | |
| Age, mean (SD) | 66.19 (13.96) | 62.96 (13.89) | <0.001 | 50.16 (18.59) | 44.56 (17.76) | <0.001 | <0.001 |
| Female (%) | 16517 (58.3) | 30266 (53.0) | <0.001 | 13894 (61.8) | 73572 (64.6) | <0.001 | <0.001 |
| Number of SBP measurements (SD) | 7.62 (6.69) | 6.03 (5.20) | <0.001 | 4.29 (4.02) | 3.90 (3.28) | <0.001 | <0.001 |
| SBP (mean) | 138.82 (14.19) | 132.26 (12.09) | <0.001 | 125.43 (14.53) | 120.07 (12.92) | <0.001 | <0.001 |
| SBP SD (mean) | 18.16 (5.14) | 7.88 (3.28) | <0.001 | 17.35 (4.53) | 6.43 (3.49) | <0.001 | <0.001 |
| **Comorbidities:** | | | | | | | |
| Diabetes Mellitus | 7919 (27.9) | 15415 (27.0) | 0.003 | 2956 (13.2) | 10575 (9.3) | <0.001 | <0.001 |
| Chronic Kidney Disease | 2192 (7.7) | 2924 (5.1) | <0.001 | 697 (3.1) | 1896 (1.7) | <0.001 | |
| Chronic Obstructive Pulmonary Disease | 3501 (12.4) | 5095 (8.9) | <0.001 | 1515 (6.7) | 4257 (3.7) | <0.001 | <0.001 |
| Dementia | 1808 (6.4) | 2217 (3.9) | <0.001 | 638 (2.8) | 1734 (1.5) | <0.001 | <0.001 |
| Depression | 6423 (22.7) | 11629 (20.4) | <0.001 | 5264 (23.4) | 24473 (21.5) | <0.001 | <0.001 |
| Parkinson's Disease | 300 (1.1) | 380 (0.7) | <0.001 | 134 (0.6) | 335 (0.3) | <0.001 | <0.001 |
| **Antihypertensive Therapy:** | | | | | | | |
| Treated with both thiazide and/or CCB AND one or more of ACEi, ARB, BB, or adrenergic antagonist therapy | 15082 (53.2) | 24943 (43.7) | <0.001 | N/A | N/A | N/A | N/A |
| Treated with one or more of ACEi, ARB, BB, or adrenergic antagonist only | 9078 (32.0) | 20401 (35.7) | <0.001 | N/A | N/A | N/A | N/A |
| Treated with thiazide and/or CCB only | 2345 (8.3) | 6202 (10.9) | <0.001 | N/A | N/A | N/A | N/A |

**High BPV denotes those with inter-visit SBP Standard Deviation exceeding 13 mm Hg**. SD = standard deviation; SBP = systolic blood pressure; CCB = calcium channel blocker; ACEi = angiotensin converting enzyme inhibitor; ARB = angiotensin receptor blocker; BB = beta blocker.

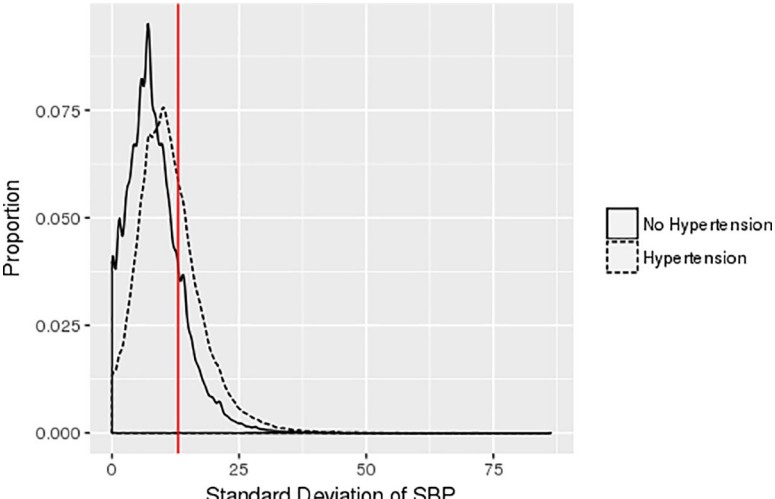

**Fig 1. Average standard deviations between visits for systolic blood pressure in patients with vs. without hypertension. Legend:** Each data point represents the mean SD for one patient's SBP between visits. The mean SD for SBP in individuals without hypertension was 8.2 and for hypertensive patients was 11.3. The red line represents a SBP SD of 13.0 mm Hg (thus values to the right of this represent patients with "high BPV").

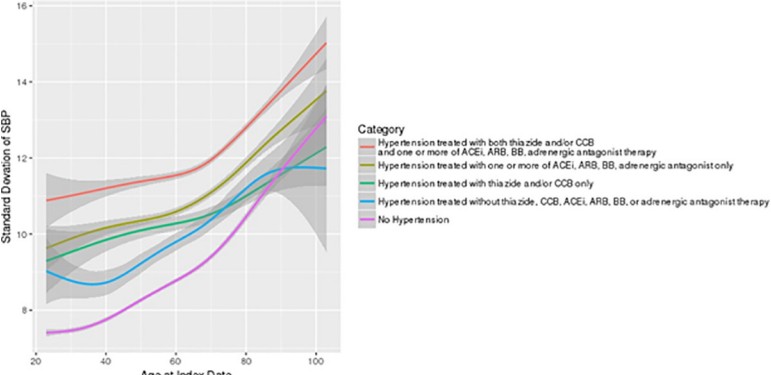

**Fig 2. Average standard deviations between visits for systolic blood pressure in patients with vs. without hypertension, by age and drug treatment.** Red: Hypertension treated with both thiazide and/or CCB and one or more of ACEi, ARB, BB, adrenergic antagonist therapy. Gold: Hypertension treated with one or more of ACEi, ARB, BB, adrenergic antagonist only. Green: Hypertension treated with thiazide and/or CCB only. Blue: Hypertension treated without thiazide, CCB, ACEi, ARB, BB, or adrenergic antagonist therapy. Purple: No Hypertension.

## Discussion

Our study adds new information in demonstrating that one sixth of non-hypertensive adults and one third of hypertensive individuals followed by primary care physicians exhibit high BPV using the SBP SD cutpoints reported in prior studies [4–9] establishing the prognostic import of BPV. Our study also demonstrates that high BPV is more common in older individuals and those with comorbidities (diabetes, chronic kidney disease, chronic obstructive pulmonary disease, dementia, depression, or Parkinson's disease). We also found that BPV was more pronounced in patients treated with ACEi or ARB and/or BB therapies than those treated with calcium channel blockers and/or thiazide diuretics. Although randomized trial data also demonstrated that the extent of BPV varies between antihypertensive drug classes, and some have posited that differential effects on BPV account for the differences between

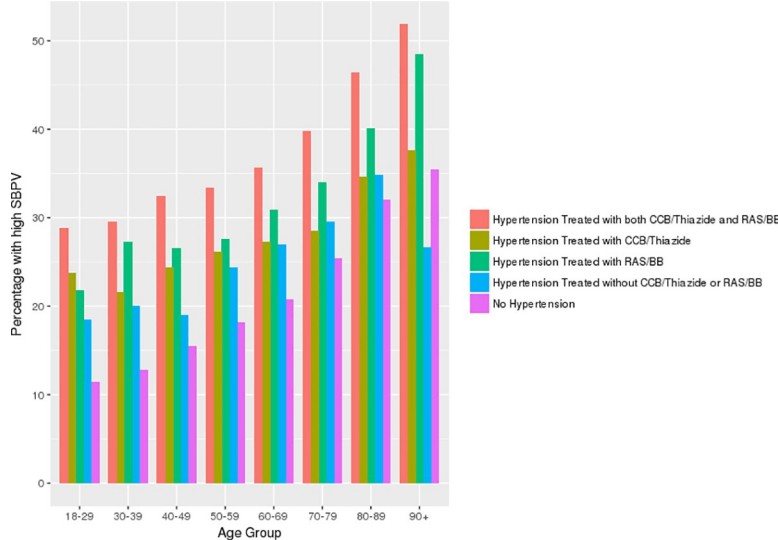

**Fig 3. Frequency of high blood pressure variability in adults, subdivided by age, hypertension status, and drug treatment.**

antihypertensive drug classes in preventing stroke [10,15], it is important to note that no studies as of yet have shown that reduction of BPV per se (independent of SBP reduction) improves outcomes [2].

Visit-to-visit BPV is a product of many factors, including the SBP range (higher mean SBP values will exhibit larger SD), the length of time between visits, the number of measures taken at each visit and the number of visits, adjustments in antihypertensive therapy or patient lifestyle, environmental conditions such as weather, and individual intrinsic characteristics such as arterial stiffness and/or altered baroreflex function [3,16,17]. While patient adherence to medications also influences BPV, medication adherence explained only a small percentage of BPV in the ALLHAT Trial [18]. While some may argue that high BPV merely reflects differences in time of day or patient anxiety levels between clinic visits, an analysis within ASCOT-PBLA demonstrated that BPV was unrelated to the white coat effect or differences in when BP was measured and that in fact visit-to-visit BPV correlated with variability in ambulatory BP monitoring results [5]. In two trial populations and a cohort of over 3 million US veterans, BPV was found to correlate strongly with the presence of arteriosclerotic risk factors such as older age, smoking, diabetes, chronic kidney disease, and peripheral arterial disease [5,8,19]. Our data confirms the association between some of these factors (older age, diabetes, and chronic kidney disease) and high BPV. Due to data limitations we could not explore the impact of smoking or peripheral arterial disease.

We acknowledge there are some limitations of our study. First, the protocol to measure SBP (for example, automated vs. manual, number of readings per visit, and physician/nurse in attendance or not) was not standardized across CPCSSN participating sites, although it was presumably consistent within each clinic and thus does not mitigate our findings on intra-individual variation in SBP levels when measured in the same clinics at different visits. Of note, the proportion of patients with high BPV was similar at all 10 participating CPCSSN sites. Second, the CPCSSN EMR data were collected for clinical (not research) purposes and thus there may be some misclassification or under-capture of comorbidities, although it seems unlikely this would have systematically differed in patients with versus without high BPV. We were unable to examine factors like smoking, alcohol use, or obesity due to high frequency of missing data in those fields, nor could we examine for comorbidities such as stroke, heart failure, or angina since they have not yet been validated within the CPCSSN database (those that have been validated were included in the Table 1) [13]. As a result we report frequencies of high BVP in individuals with specific comorbidities rather than adjusting for an incomplete list of potential confounders. Although some may argue that we should not include prescribing data in our case definition for hypertension as antihypertensives may be used for other conditions, it is worth noting that prior validation work established that our EMR-based case definition using billing codes, free text mining of progress notes, plus prescribing data in CPCSSN had a specificity of 94% [13]. Third, our antihypertensive drug data was derived from prescriptions written by CPCSSN clinicians rather than dispensations from pharmacies and thus we cannot be certain that patients were taking the medications as prescribed or examine the impact of adherence on BPV, although their high frequency of clinic visits suggests good adherence since these factors often correlate [20,21]. Fourth, while some may argue that the more times SBP is measured the lower the SD will be due to regression to the mean, it has been shown that intraindividual BPV is relatively stable after 6 measurements and our hypertensive cohort had a mean of 7 measurements each over the two years studied [22]. As hypertensive individuals had more BP measurements than those without hypertension (7 vs. 4), this in fact strengthens our confidence in our finding that hypertensive individuals are more likely to exhibit high BPV than non-hypertensives. Fifth, we do not have any data on clinical outcomes in these patients but there is already a robust evidence base establishing that high BPV

is a risk factor for cardiovascular events and even all-cause mortality [1–9,19]. Sixth, we do not have data on the presenting complaint for each clinic visit, only what the physician billed for, and thus cannot explore differences in BPV if visits were for something acute vs. routine followup monitoring. Finally, although the national CPCSSN data has a higher proportion of females and older adults than the general Canadian population and the prevalence of hypertension and other comorbidities in our study is higher than in the general Canadian population, this is reflective of a typical population attending primary care multiple times within 2 years [23].

In conclusion, the key messages from our study are that high BPV is common (one third of hypertensive and one sixth of normotensive adults), is more frequent in older adults or those with comorbidities, and appears to differ across antihypertensive treatment regimens. All of these findings suggest that BPV deserves increased attention from clinicians as visit-to-visit fluctuations in BP are often ignored or mis-interpreted as justifying clinical inertia rather than flagged as another potential risk factor in a patient warranting closer monitoring and/or intervention. Our findings also have implications for researchers: in particular, a key question for future randomized trials in hypertension management is whether differences in cardiovascular outcome rates between drug classes [24] are due to differences in their effects on BPV and whether antihypertensive treatment should be targeted towards stabilizing BPV as well as lowering mean BP levels [1].

## Author Contributions

**Conceptualization:** Finlay A. McAlister.

**Data curation:** Brendan Cord Lethebe, Alexander A. Leung, Rajdeep S. Padwal.

**Formal analysis:** Brendan Cord Lethebe, Alexander A. Leung, Rajdeep S. Padwal.

**Methodology:** Finlay A. McAlister.

**Supervision:** Finlay A. McAlister, Rajdeep S. Padwal, Tyler Williamson.

**Visualization:** Brendan Cord Lethebe.

**Writing – original draft:** Finlay A. McAlister.

**Writing – review & editing:** Finlay A. McAlister, Brendan Cord Lethebe, Alexander A. Leung, Rajdeep S. Padwal, Tyler Williamson.

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
