## [Decision Letter · Decision Letter 0]

14 Dec 2020

PONE-D-20-30322

Visit-to-visit blood pressure variability is common in primary care patients: retrospective cohort study of 221,812 adults

PLOS ONE

Dear Dr. McAlister,

Thank you for submitting your manuscript to PLOS ONE. After careful consideration, we feel that it has merit but does not fully meet PLOS ONE’s publication criteria as it currently stands. Therefore, we invite you to submit a revised version of the manuscript that addresses the points raised during the review process.

We look forward to receiving your revised manuscript.

Kind regards,

Hans-Peter Brunner-La Rocca, M.D.

Academic Editor

PLOS ONE

Journal Requirements:

2.)We note that you have indicated that data from this study are available upon request. PLOS only allows data to be available upon request if there are legal or ethical restrictions on sharing data publicly. For information on unacceptable data access restrictions, please see http://journals.plos.org/plosone/s/data-availability#loc-unacceptable-data-access-restrictions.

Reviewers' comments:

Reviewer's Responses to Questions

**Comments to the Author**

1. Is the manuscript technically sound, and do the data support the conclusions?

Reviewer #1: Yes

Reviewer #2: Yes

2. Has the statistical analysis been performed appropriately and rigorously? 

Reviewer #1: No

Reviewer #2: Yes

3. Have the authors made all data underlying the findings in their manuscript fully available?

Reviewer #1: Yes

Reviewer #2: Yes

4. Is the manuscript presented in an intelligible fashion and written in standard English?

Reviewer #1: Yes

Reviewer #2: Yes

5. Review Comments to the Author

Reviewer #1: Summary of the research and overall impression

This manuscript is a descriptive study of a large retrospective cohort of primary care patients with and without hypertension (n=221812) of the frequency of high visit-to-visit blood pressure variability, its clinical correlates and its association with antihypertensive medication. Over the course of two years high BPV was common (1/6 of non-hypertensive adults and 1/3 of hypertensive individuals), was associated with atherosclerotic comorbidities (such as diabetes, CKD, COPD) and Parkinson’s disease an dementia, and the extent of BPV varied across antihypertensive treatment regimens in a basic comparison without adjustment for other potential confounders.

Blood pressure variability, as the authors claim, is an upcoming important clinical measurement previously defined in many different ways but still an overlooked entity in clinical practice. The manuscript indeed describes the frequency of high BPV according to their definition and the conclusion correlates with the results given, but their preciseness in describing the frequency of its clinical correlates and its association with antihypertensive medication can be improved. Also all the analyses are not corrected for any confounders. In addition, the rationale behind the importance of this study is not totally clear. Also, the rationale behind the methods and presenting their results should be improved to better support their claims (descriptives) and to improve the quality of their paper.

Major issues

-In general: please specify all abbreviations, also in the tables and figures.

Introduction

1.) The introduction lacks a description of the gaps in previous studies in clinical practice and why it is so important that they fill this gap. The authors describe that the cut-off of 13 SD is based on the actual SD values of previous publication . In reference 4 of the referencelist of the article the SD value of the highest quintile is 13.66 SD. What is so different in this study?

2.) Last paragraph: the research objective/question (quantifying of the prevalence of BPV) can be described clearer.

Methods

3a.) Although it are common and relevant comorbidities, the presentation seems random, what is it based on?

3b.) Is there data on the main cardiovascular disease outcomes: peripheral arterial disease, myocardial infarction, heart failure, cerebrovascular disease; and sleep apnea as comorbidities?

3c.)The given characteristics are limited. Is there any information about the other cardiovascular risk factors: Dyslipidemia, smoking, alcohol use? Information on educational level? Since it is an electronical medical record database I would guess this information should be available.

4.)The classification of antihypertensive treatment regime seems rather random and the terms used in the table are not clear.

a).What is the reasoning behind this classification? Please describe this in the methods. Specify other therapy.

b.)What does Thiazide diuretic and/or CCB therapy with/without other therapy mean, there are so many “/”? It is a very broad description and in my opinion also includes “only thiazide diuretic and/or CCB therapy” because then it is thiazide diuretic and/or CCB therapy without other therapy. Same for ACEi/ARB and/or BB and/or adrenergic antagonist therapy with/without other therapy and only ACEi/ARB ……

I believe it would be informative to also the use of each antihypertensive class + cluster in groups (and describe why) according to treatment regime. Present the most important findings in the main manuscript and the rest in a supplemental.

5). Please provide how the blood pressure variability in SD was calculated.

6.) Statistics. Since it is a descriptive study basic statistics were used as appropriate, however it would be more informative to adjust the differences found for at least age and sex but also for other cardiovascular risk factors.

Results

7.) The main conclusion that high BPV was common in 1/6 of non-hypertensive adults and 1/3 of hypertensive individuals is in my opinion not very clear from any table of figure (you have to calculate yourself).

8.) The next conclusion is that high BPV was associated with atherosclerotic comorbidities. Without adjustment for other confounders this statement is too strong in my opinion.

Figure 1

9.) I do not think this figure in the current setting is suitable for publication. The proportions are not readable. This figure can be clearer. Where can I find that the mean SBP for nonhypertensives was 8.2 (SD 5.5) and for hypertensive people 11.3 (SD 6.3), which you mention in the main results and refer to figure 1?

Minor issues

Introduction

1.) The first paragraph reads very long, try to split it in two and be more concise (see also statement 1 of the major issues.

First part: Despite emerging evidence……….events during follow-up.[4-9] can be more concise.

Second part is then about the differences in antihypertensive treatment regimens, can also be more concise. Only discuss relevant items.

Results

2.) Hypertensive individuals also had more SBP measurements than those without a diagnosis of hypertension (mean 6.56 vs. 3.96 over the 2 years, p<0.001). How may this have influenced your results in these groups?

3.) Although systolic blood pressure variability is in clinical practice probably most relevant, did you consider diastolic blood pressure variability?

4.) Are there sex differences? Patients with hypertension and high BPV are more often female, patients without hypertension and high BPV are less often female. How do you explain this?

Discussion

5.) First paragraph: Focus first on the summary on your main findings (which is the answer on the research question/goal of the study) and then discuss the additive value to prior published work.

6.) Last paragraph: First focus on the key message, then discuss options for future research or clinical implications.

7.) Limitations. Authors should discuss the lack of adjusting their analyses for potential confounders (confounding bias) as an explanation for the differences found.

Table

8.) I miss the table legend, and abbreviation list. Provide a clear and descriptive title of the patient characteristics. The definition of high BPV should not be mentioned in the title (rather in the table legend).

9.) The given characteristics are limited. Is there any information about the other cardiovascular risk factors (see previous comment)?

Figure 2 and 3.

10.) Figures are representative but the classification of antihypertensive treatment regimes should be consistent with the classification in the table. What about the other groups?

Reviewer #2: This paper outlines the degreee of BPV in primary care patients and its correlation with comorbidities. I think this article outlines an interesting point of attention in blood-pressure management and the authors thoroughy described their research which could have important clinicl implications. However, i think some revisions should me made to increase the probability of publication.

Title: To my opinion the number of patients included in the study should not be part of the title. Perhaps it is better to remove this from the title

Abstract: The conclusion is built up from one very long sentence. Dividing this into 2 or more sentences might increase readability.

Condensed abstract: The authors divide comorbidities associated with high BPV in atherosclerotic and others. However dementia often also has a atherosclerotis etiology. Therefore, i would suggest to not make this division and just name them all comorbidities

Introduction: It might be interesting to correlate BPV with medication adherence. Is anything known about this? If not, it might be interesting to include such an analysis in your own study, if these data are available.

SD, please introduce this abbreviation when it is used the first time.

The last part of the introduction section begins with the aim of the study, but the text thereafter (starting with "we examined") to my opinion belongs to the methods section. Furthermore i would suggest to start this last section with "The aim of this study is to....

Methods: The authors used a case-definition including the use of antihypertensive drugs to identify patients with hypertension. However, the use of antihypertensive drugs does not prove that a patient has hypertension. For instance, patients with chronic heart failure, cardiovascular disease or chronic kidney disease might use AHD without having hypertension. I am not sure wheter the ause of AHD should be used to identify patients with hypertension. At least, a statement about this should be added in the limitations section.

In the part of statistical analysis it is mentioned that means and SD's versus medians and IQR's as well as the choice of the statistical test for between group comparisons was done "as appropriate". I would suggest to outline which test was used for which variables to increase readability and give better insight in the used methods.

Results: In the 10th line of this section i believe "BVP" should be changed in "BPV".

It would be interesting to know the "visit reason" and to see if this correlates with BPV> For instance an individual presenting with an COPD exacerbation (i.e. breating discomfort) or a high degree of pain might have a higher BP then in other circumstances and therefore be identified as having high BPV. I would like to suggest an extra analysis of correlation between reason of visit and BPV. If this is not possible at least something should be stated about the possible influence of reason of attendance on BPV in the discussion section.

Discussion:

In the limitations section it is stated that high frequency of clinic visits suggest a good medication adherence. I am not aware of such a correlation. However, if this correlation has already been shown in previous research, a reference should be added. Otherwise, to my opinion this statement should be removed.

To my opinion this section lacks a proper description of clinical implications of this study and BPV in general. THis should receive more attention.

6. PLOS authors have the option to publish the peer review history of their article (what does this mean?). If published, this will include your full peer review and any attached files.

Reviewer #1: No

Reviewer #2: No

---

## [Author Response · Author response to Decision Letter 0]

2 Feb 2021

January 22, 2021

Dear Dr. Brunner-La Rocca,

RE: PONE-D-20-30322

Visit-to-visit blood pressure variability is common in primary care patients: retrospective cohort study of 221,812 adults

Thank you for your email of December 14, 2020 and the opportunity to revise and resubmit our manuscript. We have uploaded both a tracked changes and a clean version of our revised manuscript and address specific reviewer comments below. 

As requested, we have included the data sharing agreement for the Canadian Primary Care Sentinel Surveillance Network (CPCSSN) – from whom we obtained the data to do this analysis (note that I had to request the data in this way and I am not the custodian of this dataset) - on page 12 of the revision as follows:

Data Sharing Agreement: Although data sharing agreements between individual practitioners and the Canadian Primary Care Sentinel Surveillance Network (CPCSSN) prohibit us from making the dataset publicly available since it contains potentially sensitive health information (a restriction imposed by the Queen’s University Health Research Ethics Board for any research use of the CPCSSN data), access may be granted to those who meet pre-specified criteria for confidential access, available by contacting info@cpcssn.org. The Canadian Primary Care Sentinel Surveillance Network (CPCSSN) has a formal data sharing policy (see http://cpcssn.ca/join-cpcssn/for-researchers/). Data will be disclosed only upon request and approval of the proposed use of the data by a review committee created by leaders of the network. This review will serve to ensure that patient privacy and rights, and data and research integrity, can be maintained. Review criteria will include demonstrated competence in data security and analysis and data will be shared to achieve the objectives in the approved protocol only. Anonymized data and a data dictionary will be made available, subject to requirements or restrictions from research ethics board or institutional review boards, existing contracts or agreements, and conditions set forth in participant consent forms. Data will be made available through secure data transfer methods overseen by CPCSSN and Queen’s University, Kingston, Ontario Canada, or by having analyses performed by the CPCSSN Staff, subject to capacity. Each proposal must identify and provide funding to defray the costs of data preparation, storage, transfer, and analysis for the organization incurring these costs.

Reviewers' comments to the Author:

Reviewer #1: 

Major issues

-In general: please specify all abbreviations, also in the tables and figures.

Reply: Done on first use of each abbreviation.

Introduction

1.) The introduction lacks a description of the gaps in previous studies in clinical practice and why it is so important that they fill this gap. The authors describe that the cut-off of 13 SD is based on the actual SD values of previous publication. In reference 4 of the reference list of the article the SD value of the highest quintile is 13.66 SD. What is so different in this study?

Reply: Done in 3rd para of the revision: “While prior publications have established that high BPV is a risk factor for subsequent cardiovascular events,[4-9] they were unable to quantify the prevalence of this risk factor since definitions were based on classifying BPV within studies on the basis of quartiles, quintiles, or deciles of SD.” Thus prior studies couldn’t define how common the problem is since the frequency was 25%, 20%, or 10% based on their method of defining “high BPV”. By pre-specifying a SD > 13 as high BPV then we could examine clinical practice EMR data to see how common high BPV was and whether it correlated with patient features or treatment patterns.

2.) Last paragraph: the research objective/question (quantifying of the prevalence of BPV) can be described clearer.

Reply: Done (see last sentence of revised 3rd para of Intro): “with the aim of examining how common high BPV is in primary care practice and whether it differs across comorbidities and treatment regimens using data from the Canadian Primary Care Sentinel Surveillance Network (CPCSSN)”

Methods

3a.) Although it are common and relevant comorbidities, the presentation seems random, what is it based on?

Reply: We have clarified that the comorbidities we examined were those that had been previously validated in CPCSSN using combinations of billing codes, progress notes, and prescription data on page 7 and we have explicitly explained at the end of that para (bottom of page 7) that “we did not extract data on conditions like stroke, heart failure, or angina for which there are no validated case definitions yet within CPCSSN.[13]”. 

3b.) Is there data on the main cardiovascular disease outcomes: peripheral arterial disease, myocardial infarction, heart failure, cerebrovascular disease; and sleep apnea as comorbidities?

Reply: No, see 3a above.

3c.)The given characteristics are limited. Is there any information about the other cardiovascular risk factors: Dyslipidemia, smoking, alcohol use? Information on educational level? Since it is an electronical medical record database I would guess this information should be available.

Reply: No, see 3a above. There was a high degree of missingness for data on alcohol use, educational level, etc in the EMR and thus we focused on conditions for which we had complete capture and pre-validated case definitions.

4.)The classification of antihypertensive treatment regime seems rather random and the terms used in the table are not clear.

a).What is the reasoning behind this classification? Please describe this in the methods. Specify other therapy.

b.)What does Thiazide diuretic and/or CCB therapy with/without other therapy mean, there are so many “/”? It is a very broad description and in my opinion also includes “only thiazide diuretic and/or CCB therapy” because then it is thiazide diuretic and/or CCB therapy without other therapy. Same for ACEi/ARB and/or BB and/or adrenergic antagonist therapy with/without other therapy and only ACEi/ARB ……I believe it would be informative to also the use of each antihypertensive class + cluster in groups (and describe why) according to treatment regime. Present the most important findings in the main manuscript and the rest in a supplemental.

Reply: We have re-classified the medications into mutually exclusive groupings to make interpretation clearer. In the second para of the Introduction we described the prior studies (refs 10 and 11) suggesting differences in BPV for patients treated with CCB or thiazides vs. those treated with ACEi, ARB, or BB and that was the basis for the groups we a priori specified.

5). Please provide how the blood pressure variability in SD was calculated.

Reply: Done (see top of page 8).

6.) Statistics. Since it is a descriptive study basic statistics were used as appropriate, however it would be more informative to adjust the differences found for at least age and sex but also for other cardiovascular risk factors.

Reply: We chose not to do incomplete adjustment and have described why on page 11: “We were unable to examine factors like smoking, alcohol use, or obesity due to high frequency of missing data in those fields, nor could we examine for comorbidities such as stroke, heart failure, or angina since they have not yet been validated within the CPCSSN database (those that have been validated were included in the Table).[13] As a result we report frequencies of high BVP in individuals with specific comorbidities rather than adjusting for an incomplete list of potential confounders.” 

Results

7.) The main conclusion that high BPV was common in 1/6 of non-hypertensive adults and 1/3 of hypertensive individuals is in my opinion not very clear from any table of figure (you have to calculate yourself).

Reply: That is why we emphasized it in the abstract and in the text at the end of para 1 in the Results section (bottom of page 8).

8.) The next conclusion is that high BPV was associated with atherosclerotic comorbidities. Without adjustment for other confounders this statement is too strong in my opinion.

Reply: Good point, and we have modified the text throughout the abstract and manuscript to instead say: “was more common in older individuals and those with comorbidities”

Figure 1

9.) I do not think this figure in the current setting is suitable for publication. The proportions are not readable. This figure can be clearer. Where can I find that the mean SBP for nonhypertensives was 8.2 (SD 5.5) and for hypertensive people 11.3 (SD 6.3), which you mention in the main results and refer to figure 1?

Reply: We have attempted to improve clarity by adding a footnote and a red vertical line to indicate the SBP SD cutpoint for high BPV. We have also added this information as text at the bottom of apge 7: ‘Patients with a diagnosis of hypertension had a mean SBP of 134.4 mm Hg with SD of 11.3 while the 136,348 patients without a diagnosis of hypertension exhibited a mean SBP of 120.9 mm Hg with SD 8.2 (p<0.001, Table).’

Minor issues

Introduction

1.) The first paragraph reads very long, try to split it in two and be more concise (see also statement 1 of the major issues.

Reply: Done

Results

2.) Hypertensive individuals also had more SBP measurements than those without a diagnosis of hypertension (mean 6.56 vs. 3.96 over the 2 years, p<0.001). How may this have influenced your results in these groups?

Reply: We have acknowledged the different frequencies of SBP and implications for our results in our limitations para on page 12: “Fourth, while some may argue that the more times SBP is measured the lower the SD will be due to regression to the mean, it has been shown that intraindividual BPV is relatively stable after 6 measurements and our hypertensive cohort had a mean of 7 measurements each over the two years studied. As hypertensive individuals had more BP measurements than those without hypertension (7 vs. 4), this in fact strengthens our confidence in our finding that hypertensive individuals are more likely to exhibit high BPV than non-hypertensives.”

3.) Although systolic blood pressure variability is in clinical practice probably most relevant, did you consider diastolic blood pressure variability?

Reply: No we did not.

Discussion

5.) First paragraph: Focus first on the summary on your main findings (which is the answer on the research question/goal of the study) and then discuss the additive value to prior published work.

Reply: Done

6.) Last paragraph: First focus on the key message, then discuss options for future research or clinical implications.

Reply: Done

7.) Limitations. Authors should discuss the lack of adjusting their analyses for potential confounders (confounding bias) as an explanation for the differences found.

Reply: Done – see page 11: “We were unable to examine factors like smoking, alcohol use, or obesity due to high frequency of missing data in those fields, nor could we examine for comorbidities such as stroke, heart failure, or angina since they have not yet been validated within the CPCSSN database (those that have been validated were included in the Table).[13] As a result we report frequencies of high BVP in individuals with specific comorbidities rather than adjusting for an incomplete list of potential confounders.”

Table

8.) I miss the table legend, and abbreviation list. Provide a clear and descriptive title of the patient characteristics. The definition of high BPV should not be mentioned in the title (rather in the table legend).

Reply: Done

9.) The given characteristics are limited. Is there any information about the other cardiovascular risk factors (see previous comment)?

Reply: No (see answer to 3a) – we have added this as a limitation on page 11. 

Figure 2 and 3.

10.) Figures are representative but the classification of antihypertensive treatment regimes should be consistent with the classification in the table. 

Reply: Done

Reviewer #2: 

Title: To my opinion the number of patients included in the study should not be part of the title. Perhaps it is better to remove this from the title

Reply: We left the size of the cohort in the title so that those doing a systematic review of the topic in the future would know that the sample size is sufficiently large that it would be worth pulling the full paper for examination.

Abstract: The conclusion is built up from one very long sentence. Dividing this into 2 or more sentences might increase readability.

Reply: Done

Condensed abstract: The authors divide comorbidities associated with high BPV in atherosclerotic and others. However dementia often also has a atherosclerotis etiology. Therefore, i would suggest to not make this division and just name them all comorbidities

Reply: Done

Introduction: It might be interesting to correlate BPV with medication adherence. Is anything known about this? If not, it might be interesting to include such an analysis in your own study, if these data are available.

Reply: We agree this would be an interesting analysis but we only have prescribing data, not data on medication adherence, in this EMR and have added that limitation in the middle of page 12.

SD, please introduce this abbreviation when it is used the first time.

Reply: Done

The last part of the introduction section begins with the aim of the study, but the text thereafter (starting with "we examined") to my opinion belongs to the methods section. Furthermore i would suggest to start this last section with "The aim of this study is to....

Reply: Done, although we went with the wording: “with the aim of examining how common high BPV is in primary care practice and whether it correlates with comorbidities and treatment regimens using data from the Canadian Primary Care Sentinel Surveillance Network (CPCSSN)”

Methods: The authors used a case-definition including the use of antihypertensive drugs to identify patients with hypertension. However, the use of antihypertensive drugs does not prove that a patient has hypertension. For instance, patients with chronic heart failure, cardiovascular disease or chronic kidney disease might use AHD without having hypertension. I am not sure wheter the ause of AHD should be used to identify patients with hypertension. At least, a statement about this should be added in the limitations section.

Reply: We have added a sentence to the limitations section on page 11: “Although some may argue that we should not include prescribing data in our case definition for hypertension as antihypertensives may be used for other conditions, it is worth noting that prior validation work established that our EMR-based case definition using billing codes, free text mining of progress notes, plus prescribing data in CPCSSN had a specificity of 94%.[13]”

In the part of statistical analysis it is mentioned that means and SD's versus medians and IQR's as well as the choice of the statistical test for between group comparisons was done "as appropriate". I would suggest to outline which test was used for which variables to increase readability and give better insight in the used methods.

Reply: Done

Results: In the 10th line of this section i believe "BVP" should be changed in "BPV".

Reply: Thank you for catching that typo, now corrected.

It would be interesting to know the "visit reason" and to see if this correlates with BPV> For instance an individual presenting with an COPD exacerbation (i.e. breating discomfort) or a high degree of pain might have a higher BP then in other circumstances and therefore be identified as having high BPV. I would like to suggest an extra analysis of correlation between reason of visit and BPV. If this is not possible at least something should be stated about the possible influence of reason of attendance on BPV in the discussion section.

Reply: We agree this would be an interesting analysis but we do not have data on presenting complaint, and have added this limitation on page 12: “Sixth, we do not have data on the presenting complaint for each clinic visit, only what the physician billed for, and thus cannot explore differences in BPV if visits were for something acute vs. routine followup monitoring.”

Discussion:

In the limitations section it is stated that high frequency of clinic visits suggest a good medication adherence. I am not aware of such a correlation. However, if this correlation has already been shown in previous research, a reference should be added. Otherwise, to my opinion this statement should be removed.

Reply: Done (refs 20 and 21 added).

To my opinion this section lacks a proper description of clinical implications of this study and BPV in general. THis should receive more attention.

Reply: Done – we added the sentence: “All of these findings suggest that BPV deserves increased attention from clinicians as visit-to-visit fluctuations in BP are often ignored or mis-interpreted as justifying clinical inertia rather than flagged as another potential risk factor in a patient warranting closer monitoring and/or intervention.”

We thank the reviewers for their helpful and constructive suggestions to improve our manuscript. Thank you for your continued consideration of our work.

Sincerely,

Finlay McAlister on behalf of all co-authors

---

## [Decision Letter · Decision Letter 1]

25 Feb 2021

Visit-to-visit blood pressure variability is common in primary care patients: retrospective cohort study of 221,803 adults

PONE-D-20-30322R1

Dear Dr. McAlister,

We’re pleased to inform you that your manuscript has been judged scientifically suitable for publication and will be formally accepted for publication once it meets all outstanding technical requirements.

Kind regards,

Hans-Peter Brunner-La Rocca, M.D.

Academic Editor

PLOS ONE

Additional Editor Comments (optional):

Reviewers' comments:

Reviewer's Responses to Questions

**Comments to the Author**

1. If the authors have adequately addressed your comments raised in a previous round of review and you feel that this manuscript is now acceptable for publication, you may indicate that here to bypass the “Comments to the Author” section, enter your conflict of interest statement in the “Confidential to Editor” section, and submit your "Accept" recommendation.

Reviewer #1: All comments have been addressed

2. Is the manuscript technically sound, and do the data support the conclusions?

Reviewer #1: Yes

3. Has the statistical analysis been performed appropriately and rigorously? 

Reviewer #1: Yes

4. Have the authors made all data underlying the findings in their manuscript fully available?

Reviewer #1: Yes

5. Is the manuscript presented in an intelligible fashion and written in standard English?

Reviewer #1: Yes

6. Review Comments to the Author

Reviewer #1: Summary of the research and overall impression

This manuscript is a descriptive study of a large retrospective cohort of primary care patients with and without hypertension (n=221812) of the frequency of high visit-to-visit blood pressure variability, its clinical correlates and its association with antihypertensive medication. Over the course of two years high BPV was common (1/6 of non-hypertensive adults and 1/3 of hypertensive individuals), was associated with comorbidities (such as diabetes, CKD, COPD, Parkinson’s disease and dementia). and the extent of BPV varied across antihypertensive treatment regimens in a basic comparison.

The authors have improved the rationale behind the importance of this study and also behind the methods and presenting the results. The quality of the paper has improved. Unfortunately no data were available on other important cardiovascular comorbidities and classical complications and risk factors (yet) , but the author's addressed this limitation accordingly in the discussion.

One minor comment: remove the term 'atherosclerotic' in the conclusion of the abstract (because you have removed it in the rest of the manuscript)

7. PLOS authors have the option to publish the peer review history of their article (what does this mean?). If published, this will include your full peer review and any attached files.

Reviewer #1: No

---

## [Editor Report · Acceptance letter]

2 Mar 2021

PONE-D-20-30322R1 

Visit-to-visit blood pressure variability is common in primary care patients: retrospective cohort study of 221,803 adults 

Dear Dr. McAlister:

I'm pleased to inform you that your manuscript has been deemed suitable for publication in PLOS ONE. Congratulations! Your manuscript is now with our production department. 

Kind regards, 

on behalf of

Dr. Hans-Peter Brunner-La Rocca 

Academic Editor

PLOS ONE